



# Stabilized two-phase material point method for hydromechanical coupling problems in solid-fluid porous media

Xiong Tang[1, 2, 3], Wei Liu[1, 2], Siming He[1, 2], Lei Zhu[1, 2], Michel Jaboyedoff[3],

Huanhuan Zhang[1, 2], Yuqing Sun[1, 2], Zenan Huo[3]

[1] Institute of Mountain Hazards and Environment, Chinese Academy of Sciences,

Chengdu 610041, China

[2] University of Chinese Academy of Sciences, Beijing 100049, China

[3] Analysis Group, Institute of Earth Sciences, University of Lausanne, CH 1015

Lausanne, Switzerland

Correspondence: Wei Liu (spon@imde.ac.cn), Siming He (hsm@imde.ac.cn)

**Abstract.** For the hydromechanically coupling of solid-fluid porous media, this study presents an explicit stabilized two phase MPM formulation based on the one-point two-phase MPM scheme. To mitigate the spurious pore pressure and maintain the numerical stability, the stabilized techniques including the strain smoothing method and the multi-field variational principle are implemented in the proposed formulation. The strain smoothing technique is used to smooth the volumetric strain rate, and the calculation of the pore pressure increasement at particles is based on the multi-field variational principle. Four numerical examples are performed to evaluate the performance of the proposed formulation. With its effective and easy implemented stabilized techniques, the proposed formulation provides stable and reliable outcomes that well align with analytical solutions and results from other approaches, offering extensively validation that the proposed two phase MPM formulation is an effective and reliable approach for the simulation of solid-fluid porous media under both static and dynamic conditions.

## 1 Introduction

The hydromechanically coupling of solid-fluid porous media widely presents in



nature and engineering, from natural processes like rainfall-induced landslide and
earthquake-induced liquefaction, to coastal dike-breaking and offshore foundations
(Jerolmack and Daniels, 2019; Zhan et al., 2025; Guan and Shi, 2023). Due to the
practical importance, reproducing and understanding the physical nature of such a
two-phase system have attracted strong research interests across many scientific and
engineering disciplines, which has become increasingly recognized with recent
advances in both observational and simulation tools(Li et al., 2023; Taylor-Noonan et
al., 2022; Pudasaini and Mergili, 2019). While the numerical modeling of this two-
phase coupling system is still a significant challenge for researchers in many
disciplines alike.
In soil-fluid coupling problems, the motion of each constituent is governed by stress
distributions, external gravity forces and interaction forces (Pudasaini and Mergili,
2019; Baumgarten and Kamrin, 2018; Bandara and Soga, 2015). To better simulate
this two-phase system, various numerical methods have been proposed, including the
smoothed particle hydrodynamics (SPH) method (Lian et al., 2023; Chen et al., 2023),
the particle finite element method (Yuan et al., 2022; Jin and Yin, 2022), and the
material point method (MPM) (Bandara and Soga, 2015; Bandara et al., 2016; Jassim
et al., 2013; Yerro et al., 2015; Wyser et al., 2020). Among these methods, MPM has
been shown to be both useful and efficient for simulating large deformation problems
with history-dependent materials. Originated from the particle-in-cell (PIC) method,
MPM is a hybrid Euler-Lagrangian method which has significant advantages in
dealing with large deformation problems (Li et al., 2020; Zhao et al., 2023; Fernández
et al., 2023). In MPM, a continuum body is discretized by a group of material points
carrying all physical information like displacement, velocity, stress, strain, etc. At
each time step, the physical information at particles is interpolated to the background
mesh node, which is essentially a Eulerian mesh, and then the governing equations
can be solved on it. Subsequently, the solution is re-interpolated to each material
particle to update the physical information. The original background mesh can be used
again in the new time step, which can eliminate the mesh distortion problem in
Lagrangian method, and the accuracy of the solution for large deformation problem





can be guaranteed (Fig. 1). Currently, various coupling MPM formulations have been
proposed (i.e. the one-point or two-point schemes (Bandara and Soga, 2015; Jassim et
al., 2013), the solid displacement-fluid pressure or solid velocity-fluid velocity
formulation (Zhang et al., 2009; Lei et al., 2020)) and have been widely used in two-
phase coupling problems and engineering applications (Du et al., 2023; Ceccato et al.,
2024; Shen et al., 2024; Zheng et al., 2024a; Yamaguchi et al., 2023; Zheng et al.,
2024b; Zhan et al., 2025).

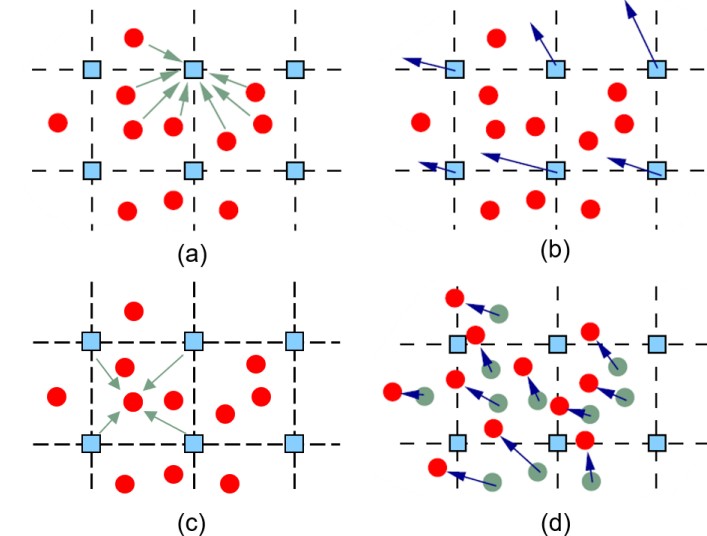

**Figure 1.** Standard algorithm of MPM: (a) interpolating information from particles to
nodes; (b) solving governing equations on nodes; (c) interpolating information from
nodes to particles; (d) update particles information.
However, the standard MPM formulation usually employs low-order shape functions
within an explicit time integration scheme for simplicity and efficiency, which suffers
from the cell-crossing error and the volumetric locking when applied to coupled
hydromechanical problems (Li et al., 2024; Sang et al., 2024). The cell-crossing error
during particle movement arises from the use of low-order shape functions, which
have discontinuous gradients between background cells. To address this issue, higher-
order interpolation functions with continuous gradients across elements can be
employed, such as the Generalized Interpolation Material Point (GIMP) method



(Bardenhagen and Kober, 2004), the B-spline method (De Vaucorbeil et al., 2020) and
the Convected Particle Domain Interpolation (CPDI) (Wang et al., 2023b). Due to the
low compressibility of pore fluid and limited permeability, the high stiffness of the
pore fluid and low permeability will lead to volumetric locking and erroneous strain,
which may not only result in undesired pore pressure oscillation, but also render the
simulation highly unstable. Various numerical stabilization techniques have been
implemented in MPM to solve this issue, including the reduce integration (Bandara
and Soga, 2015; Zheng et al., 2021), the B-bar approach (Wang et al., 2018; Tang et
al., 2024), the nodal or cell smoothing method (Lei et al., 2020; Wang et al., 2023a),
the fractional stepping method (Kularathna et al., 2021; Jassim et al., 2013), the
polynomial pressure projection method (Zhao and Choo, 2020), the multi-field
variational principle (Liu et al., 2020; Zheng et al., 2021; Tang et al., 2024; Zheng et
al., 2022), and coupling with other algorithms (Baumgarten et al., 2021; Li et al., 2024;
Tran et al., 2023; Sang et al., 2024). Although these techniques produce results that
overcome volumetric locking and reduce pore pressure oscillation, some are
conditionally stable, and some require significant modifications of the existing MPM
algorithm, leading to additional computation cost and difficulty (Lei et al., 2020; Li et
al., 2024). Therefore, their usage should depend on the specific problem at hand.
More the features and limitations of these techniques can be found in the summary of
Li et al. (Li et al., 2024) and Sang et al. (Sang et al., 2024).
Here, based on the one-point two-phase MPM scheme (Jassim et al., 2013), we
proposes an explicit stabilized two-phase MPM formulation for both static and
dynamic analyses of solid-fluid porous media. To avert the volumetric locking and
maintain the numerical stability, the stabilized techniques including the strain
smoothing method (Mast et al., 2012) and the multi-field variational principle (Chen
et al., 2018) have been implemented in the proposed formulation. The strain
smoothing method is employed to smooth the volumetric strain rate, and the
calculation of the pore pressure increasement at particles is based on the multi-field
variational principle for accuracy and stability. The spurious pore pressure oscillation
can be well mitigated during pore pressure calculation and interpolation. With these



effective and easy implemented techniques, the volumetric locking can be
significantly eliminated under both static and dynamic conditions. The study is
organized as follows. First, the governing equations for two-phase flow are briefly
introduced in Section 2. The numerical implementation of the proposed formulation
and the stabilized techniques are presented in section 3. And then four numerical
examples for the verification of the proposed method are performed and analyzed in
section 4. Finally, discussion and conclusion are drawn in the last section.
**2 Governing equations**

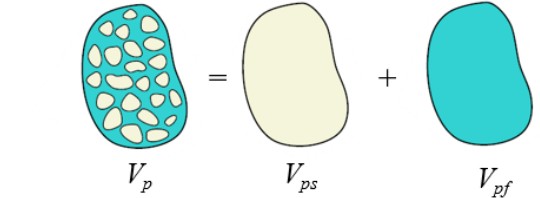

**Figure 2.** Sketch of material point composition in single-point-two-phase MPM
model (Kularathna et al., 2021).
In one-point two-phase MPM formulation, according to the theory of mixture
(Baumgarten and Kamrin, 2018), the representative volume (RVE) $V_p$ of a particle
material particle is a summation of solid phase volume $V_{sp}$ and fluid phase volume $V_{fp}$,
and each phase (solid, fluid) in the RVE can be characterized by its volume fraction
(Fig. 2). The apparent density of each phase is characterized by the intrinsic density
with the volume fraction, which reads,
$$\bar{\rho}_s = \varphi \rho_s, \ \bar{\rho}_f = n\rho_f \tag{1}$$
where $\varphi$ is the solid volume fraction, $n$ is the porosity, $\rho_s$ and $\rho_f$ are the intrinsic
density of solid and fluid, respectively; $\bar{\rho}_s$ and $\bar{\rho}_f$ are the apparent density of solid
and fluid, respectively.
**2.1 Mass conservation equations**
The mass conservations in a part of the solid/fluid phase continuum in Lagrangian



description are expressed as,
$$\frac{D^s \bar{\rho}_s}{Dt} + \bar{\rho}_s \nabla \cdot \mathbf{v}_s = 0 \tag{2}$$

$$\frac{D^f \bar{\rho}_f}{Dt} + \bar{\rho}_f \nabla \cdot \mathbf{v}_f = 0 \tag{3}$$

where $\mathbf{v}_s$, $\mathbf{v}_f$ are the velocity of solid and fluid phases in their reference frame,
respectively. In microscale, the solid grain is assumed to be incompressible, so $\rho_s$ is
constant. However, $\bar{\rho}_s$ will change when the solid phase compacts or dilates due to
the deformation of the solid skeleton structure. Therefore, a simple expansion of Eq.
(2) using the definition of porosity yields an expression for the change rate of the local
measure of porosity,
$$\frac{D^s n}{Dt} = (1-n)\nabla \cdot \mathbf{v}_s \tag{4}$$

In one-point two-phase MPM formulation, all constituents are represented by the
same Lagrangian material point in the current configuration. The material time
derivative of the fluid phase with respect to the motion of the solid phase is described
as follows,
$$\frac{D^f}{Dt} = \frac{D^s}{Dt} + (\mathbf{v}_f - \mathbf{v}_s) \cdot \nabla \tag{5}$$

So, Eq. (3) can be expressed as,
$$\frac{D^s \bar{\rho}_f}{Dt} + (\mathbf{v}_f - \mathbf{v}_s) \cdot \nabla \bar{\rho}_f + \bar{\rho}_f \nabla \cdot \mathbf{v}_f = 0 \tag{6}$$

And Eq. (6) can be further written as,
$$n\frac{D^s \rho_f}{Dt} + \rho_f \frac{D^s n}{Dt} + (\mathbf{v}_f - \mathbf{v}_s) \cdot \nabla n\rho_f + n\rho_f \nabla \cdot \mathbf{v}_f = 0 \tag{7}$$

Assuming the fluid phase is barotropic, density variation in a barotropic fluid obeys
the following relationship,
$$\frac{1}{\rho_f}\frac{D^s \rho_f}{Dt} = \frac{1}{K_f}\frac{D^s p_f}{Dt} \tag{8}$$

where $K_f$ is the bulk modulus of fluid, $p_f$ is the pore fluid pressure.
Combining with Eq. (4) and neglecting spatial variations in density and porosity, the
pore pressure change rate can be obtained,
$$\frac{D^s p_f}{Dt} = -\frac{K_f}{n}[(1-n)\nabla \cdot \mathbf{v}_s + n\nabla \cdot \mathbf{v}_f] \tag{9}$$





## 2.2 Momentum conservation equations

The momentum conservation equations for each continuum phase are given as,

$$\bar{\rho}_s \frac{D^s \mathbf{v}_s}{Dt} = \bar{\rho}_s \mathbf{b} - f_b - f_d + \nabla \cdot \boldsymbol{\sigma}_s \qquad (10)$$

$$\bar{\rho}_f \frac{D^f \mathbf{v}_f}{Dt} = \bar{\rho}_f \mathbf{b} + f_b + f_d + \nabla \cdot \boldsymbol{\sigma}_f \qquad (11)$$

where $\mathbf{b}$ is the body force, which is equal to the gravitational acceleration; $f_b$ and $f_d$ are the buoyant force and inter-phase body force, respectively; $\boldsymbol{\sigma}_s$ and $\boldsymbol{\sigma}_f$ are the solid and fluid stress, respectively. Due to the viscous effects, a flow through porous media results in a drag force, which can be considered as a body force enforced on one phase from the other phase. The classic Darcy's law describes a linear drag force as,

$$f_d = \frac{n \bar{\rho}_f g}{K_s} (\mathbf{v}_s - \mathbf{v}_f) \qquad (12)$$

where $K_s$, in the unit of m/s, is the hydraulic conductivity ($K_s = \rho_f g k / \mu_f$, where $k$ is intrinsic permeability in the unit of m$^2$ and $\mu_f$ is the dynamic viscosity of fluid). This linear relation has been employed in several studies (Zhan et al., 2023; Liu et al., 2017) to model the drag force in saturated porous media when the pore flows are in the laminar flow range with a relatively low Reynolds number. While, the buoyant force, $f_b$, which yields the form for immiscible mixtures,

$$f_b = p_f \nabla n \qquad (13)$$

And the solid phase stress $\boldsymbol{\sigma}_s$ is taken following the effective stress classic form,

$$\boldsymbol{\sigma}_s = \boldsymbol{\sigma}'_s - (1-n) p_f \mathbf{I} \qquad (14)$$

where $\mathbf{I}$ is a 3×3 identity matrix, $\boldsymbol{\sigma}'_s$ is the effective solid phase related to the deformation of the solid phase matrix, which excludes the pressurization of the solid phase due to the pressure of the pore fluid. And the fluid phase stress $\boldsymbol{\sigma}_f$ is simplified into an isotropic pressure, $np_f\mathbf{I}$, which is expressed as,

$$\boldsymbol{\sigma}_f = -np_f \mathbf{I} \qquad (15)$$

Finally, the momentum equations for solid and fluid phase are given as,

$$\bar{\rho}_s \frac{D^s \mathbf{v}_s}{Dt} = \bar{\rho}_s g - f_d + \nabla \cdot \boldsymbol{\sigma}'_s - (1-n)\nabla p_f \qquad (16)$$



$$\bar{\rho}_f \frac{D^f \mathbf{v}_f}{Dt} = \bar{\rho}_f g + f_d - n\nabla p_f \tag{17}$$

With a proper constitutive rule governing the mechanical behavior of the solid
effective stress $\boldsymbol{\sigma}'_s$, the equations can fully capture the motion and physical behavior
of this two-phase system.
**3 Numerical implementations**
**3.1 Discretized of governing equations**
In MPM, the material domain is discretized into Lagrangian material points under
Euler background mesh. The field variables of particles can be interpolated to the
background mesh nodes through shape functions. For instance, the displacement and
its derivative at particle $p$ is expressed as,
$$u_{pi} = \sum_{I=1}^{N_g} N_{Ip} u_{Ii} \tag{18}$$

$$u_{pi,j} = \sum_{I=1}^{N_g} N_{Ip,j} u_{Ii} \tag{19}$$

where subscripts $i$ and $j$ denote the components of tensor, which follow the Einstein
summation convention, and comma between the subscripts indicates partial
derivatives; $u_{Ii}$ is the displacement at grid node $I$, $N_{Ip} = N_I(x_p)$ is the shape function of
particle $p$ at grid node $I$, $x_p$ denotes the coordinates of particle $p$, $N_{Ip,j}$ is the derivative
of shape functions, $N_g$ is total the grid node number. In this study, the GIMP shape
function (Bardenhagen and Kober, 2004) and discretization is used to avoid the stress
oscillation promoted by the cell-crossing error.
By this way, the momentum equations are discretized in space by means of the
Galerkin method considering nodal shape functions. And a discretized form of
momentum equation of soil Eq. (16) on mesh node is expressed as,
$$m_{sI} a_{sIi} = f_{sIi}^{int} + f_{sIi}^{ext} \tag{20}$$

where $m_{sI} = \sum_{p=1}^{N_p} N_{Ip} m_{sp}$ is the node mass for solid, in which $N_p$ is total the number of
particles and $m_{sp}$ is the particle solid mass; $a_{sIi}$ is the solid acceleration at node, $f_{sIi}^{int}$



and $f_{sIi}^{ext}$ are the internal and external nodal forces, respectively.
The internal nodal force is expressed as,

$$f_{sII}^{int} = \sum_{p=1}^{N_p}(1-n_p)N_{Ip,j}p_{fp}V_p - \sum_{p=1}^{N_p}N_{Ip,j}\sigma'_{spij}V_p \qquad (21)$$

where $\sigma'_{spij}$ is the effective stress of material particle $p$, $p_{fp}$ is the pore pressure of
material particle, $n_p$ is the material particle porosity, $V_p$ is the volume of material
particle $p$.
The external grid nodal force is expressed as,

$$f_{sII}^{ext} = \sum_{p=1}^{N_p}N_{Ip}m_{sp}b_i - \sum_{p=1}^{N_p}N_{Ip}f_dV_p + \int_{\partial\Omega}N_{Ip}\overline{\mathbf{T}}_s dS - \int_{\partial\Omega}(1-n_p)N_{Ip}\overline{\mathbf{P}}dS \quad (22)$$

where $\overline{\mathbf{T}}_s$ and $\overline{\mathbf{P}}$ are the prescribed traction and the prescribed pressure on the
boundary $\partial\Omega$, respectively; $dS$ denotes the surface integral that is only non-zero at the
boundary $\partial\Omega$.
Likewise, a discretized form of the momentum equation of fluid Eq. (17) on the mesh
node can be expressed as,

$$m_{fl}a_{fli} = f_{fli}^{int} + f_{fli}^{ext} \qquad (23)$$

where $m_{fl} = \sum_{p=1}^{N_p}m_{fp}N_{Ip}$ the grid node mass for fluid, in which $m_{fp}$ is the particle fluid
mass; $f_{fli}^{int} = \sum_{p=1}^{N_p}n_pN_{Ip,j}p_{fp}V_p$ represents the nodal internal force from pore pressure
gradient, $f_{fli}^{ext} = \sum_{p=1}^{N_p}N_{Ip}m_{sp}b_i + \sum_{p=1}^{N_p}N_{Ip}f_dV_p - \int_{\partial\Omega}n_pN_{Ip}\overline{\mathbf{P}}dS$ denotes the nodal external forces
from body force, inter-phase drag force and the boundary prescribed pressure, $a_{fli}$ is
the fluid phase acceleration at mesh node, $b_i$ is the body force vector.
Meanwhile, the strain rate associated with the material point is calculated with its
corresponding nodal velocity,

$$\dot{\varepsilon}_{spij} = \sum_{I=1}^{n_g}[N_{Ip,j}v_{si} + (N_{Ip,j}v_{si})^T]/2 \qquad (24)$$

$$\dot{\varepsilon}_{fpij} = \sum_{I=1}^{n_g}[N_{Ip,j}v_{fi} + (N_{Ip,j}v_{fi})^T]/2 \qquad (25)$$





where $v_{si}$ and $v_{fi}$ are the nodal velocity for the solid phase and fluid phase, respectively;
$\dot{\varepsilon}_{spij}$ and $\dot{\varepsilon}_{fpij}$ are the particle strain rate for the solid phase and fluid phase, respectively.

### 3.2 Numerical stability

As mentioned above, the solid-fluid coupling MPM suffers from the volumetric
locking. The stabilized technique is needed for the stability of the simulation. Here, to
mitigate the pore pressure oscillation and maintain the numerical stability, the strain
smoothing method is used to smooth the particle volumetric strain rate, while the pore
pressure increasement at particles is calculated based on the multi-field variational
principle for the stability, accuracy and smoothness of the results.

### 3.2.1 Strain smoothing method

The numerically stress/strain smoothing method has been used in the two-phase
saturated and unsaturated MPM formulations (Lei et al., 2020; Wang et al., 2023a)
and can effectively mitigate the stress oscillation in a simple way. Here, for simplicity
and efficiency, a cell-based average approach (Mast et al., 2012) is employed to
smooth the particle volumetric strain rate. By doing this, the volumetric strain rate of
material points $p$ is replaced by the averaged field value of the cell $c$ which it belongs,

$$\alpha_p = \sum_{p \in c} \alpha_p m_p / \sum_{p \in c} m_p \qquad (26)$$

where $\alpha_p$ represents the variables include the volumetric strain rate of solid and fluid,
$m_p$ is the mass of material point, representing the solid or fluid mass in different
phases.
From the averaged volumetric strain rates $\overline{\varepsilon}_v$, the updated strain rates $\tilde{\dot{\varepsilon}}_{ij}$ is computed
by means of,

$$\tilde{\dot{\varepsilon}}_{ij} = \dot{\varepsilon}_d + \overline{\varepsilon}_v \delta_{ij} / 3 \qquad (27)$$

where $\dot{\varepsilon}_d$ is the deviatoric strain rate, $\delta_{ij}$ is the Kronecker delta. On the basis of the
modified strain rates, stresses can be directly computed using the constitutive relation.

### 3.2.2 The multi-field variational principle



Since the formulation of MPM is similar to the traditional FEM, the similar
techniques used in FEM for volumetric locking are also applicable to the MPM. The
multi-field variational principle is a commonly used anti-locking technique in the
finite element method (FEM) without using high-order shape functions. In MPM,
Chen et al. (Chen et al., 2018) first used the multi-field variational principles to
mitigate volumetric-locking and numerical oscillation in weakly compressible
problems. And then Liu et al. (Liu et al., 2020) and Tang et al. (Tang et al., 2024)
applied this technique in the sing-point two phase unsaturated MPM formulation to
mitigate volumetric-locking and carried out the simulation of the Hong Kong Tsui
Load landslide and Yanyuan landslide. Zheng et al. (Zheng et al., 2021, 2022) used
the multi-field variational principles for the patch recovery of pore pressure increment
in the explicit two-point two phase MPM formulation and fully implicit MPM
formulation. Based on the multi-field variational principle, the pore pressure field is
approximated by expressing the pore pressure increment and the test function as
(Chen et al., 2018),
$$\dot{p}_f(\mathbf{x},t) = Q^T(\mathbf{x})a(t) \tag{28}$$

$$\delta\dot{p}_f(\mathbf{x},t) = \delta a^T(t)Q(\mathbf{x}) \tag{29}$$

where $\boldsymbol{Q}$ and $\boldsymbol{a}$ are the polynomial basis function and coefficient vector to be solved.
The polynomial basis function can be constant, linear, or quadratic (i.e., $\boldsymbol{Q}$ = [1], [1, $x$,
$y$, $z$], or [1, $x$, $y$, $z$, $x^2$, $xy$, $y^2$, $yz$, $z^2$, $zx$], and the corresponding coefficient $a$ = [$a_0$], [$a_0$,
$a_1$, $a_2$, $a_3$]$^T$, or [$a_0$, $a_1$, $a_2$, $a_3$, $a_4$, $a_5$, $a_6$, $a_7$, $a_8$, $a_9$, $a_{10}$]$^T$). Here, in the single-point two-
phase MPM formulation, the weak form of the pore pressure rate can be expressed as,
$$\int_\Omega \delta\dot{p}_f(\dot{p}_f + \frac{K_f}{n}[(1-n)\nabla\cdot\mathbf{v}_s + n\nabla\cdot\mathbf{v}_s]d\Omega = 0 \tag{30}$$

And then, the weak form can be changed to,
$$\int_\Omega Q\frac{K_f}{n}[(1-n)\nabla\cdot\mathbf{v}_s + n\nabla\cdot\mathbf{v}_s]d\Omega = -a\int_\Omega QQ^T d\Omega \tag{31}$$

The coefficient can be further expressed as,
$$a = -\mathbf{H}^{-1}\int_\Omega Q\frac{K_w}{n}[(1-n)\nabla\cdot\mathbf{v}_s + n\nabla\cdot\mathbf{v}_s]d\Omega \tag{32}$$



where $\mathbf{H} = \int_{\Omega} QQ^T d\Omega$. In order to solve the coefficient vector, the node-based method
(Mast et al., 2012) is used due to its simplicity and efficiency. Using the node-based
method, the node coefficient vector is written as,

$$\mathbf{a}_I = -\mathbf{H}_I^{-1} \sum_{p=1}^{n_p} N_{Ip} Q_p \frac{K_f}{n} [(1-n)\nabla \cdot \mathbf{v}_s + n\nabla \cdot \mathbf{v}_s] V_p \tag{33}$$

where $\mathbf{H}_I = \sum_{p=1}^{n_p} Q_p Q_p^T N_{Ip} V_p$. After solving the coefficient vector for each node, the
changing rate of pore pressure can be written as

$$\dot{p}_{fp} = Q_p^T \sum_{I=1}^{n_g} a_I N_{Ip} \tag{34}$$

where $\sum_{I=1}^{n_g} a_I N_{Ip}$ is the node value interpolated to the particle.
**3.3 Numerical algorithm**





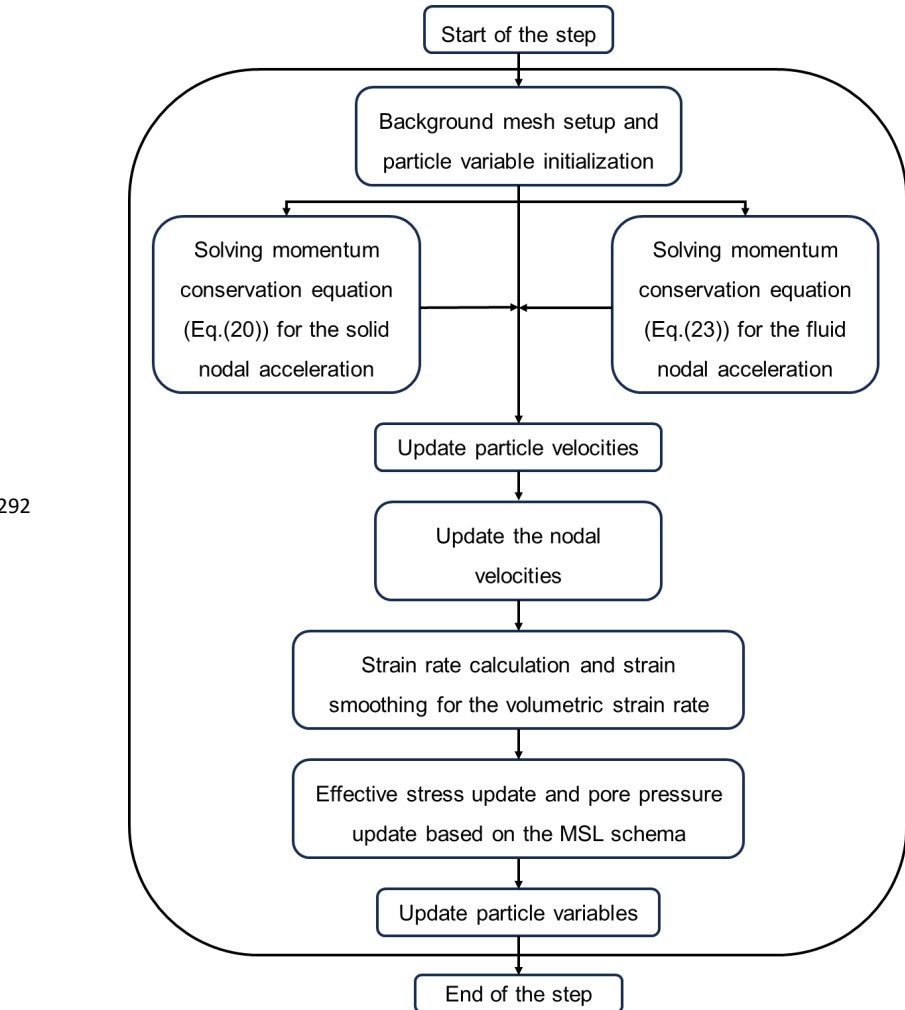


**Figure 3.** Numerical implementation procedure of the proposed stabilized two phase

294                                         MPM formulation.

In the proposed formulation, each time step is solved explicitly according to the
following sequence of sub-steps (see Fig. 3):
(1) All the variables associated with each material point are initialized first (initial
position, stress, pore pressure, etc.);
(2) Interpolate the variables of material points to the nodes of the background mesh
using the shape function calculated based on particle locations with respect to the
background mesh nodes;



(3) Combined with the correct boundary conditions, the accelerations of each phase
on the background mesh node are obtained based on Eq. (20) and (23);
(4) Update the velocity of all material points for both phases using the FLIP scheme
(Hammerquist and Nairn, 2017);
(5) Update the nodal velocities for both phases by interpolating velocities back from
the material points;
(6) Strain rate increments of solid and fluid phase on particles are calculated, and the
cell-based strain smoothing technique expressed in Eq. (26) is applied to smooth the
volumetric strain rate;
(7) Update the effective stress based on its constitutive model and the pore pressure
based on the multi-field variational principle;
(8) Update the state variables at particles, such as particle volume, porosity and
position;
(9) Reset the background mesh for the next step and store all the updated information
in material points.

## 4 Numerical examples

In this section, four numerical examples are provided to demonstrate the performance
of the proposed MPM formulation. First, a one-dimensional consolidation under both
small and large conditions is simulated. Subsequently, the two-dimensional
consolidation under local loading and the cyclic loading test are conducted to show its
efficacy under external loading. And then, the self-wight consolidation is analyzed to
illustrate its capability in simulating undrained and drained conditions, as well as large
deformation situation.

### 4.1 One-dimensional consolidation

The one-dimensional consolidation problem has been frequently studied to verify and
assess numerical methods, as it allows a direct comparison with analytical solutions.
Here, both small and large deformation conditions are conducted and the numerical
results are compared with their corresponding analytical solutions.





### 4.1.1 Small deformation

As shown in Fig. 4, a saturated soil column with a width of 0.1 m and a length of 1.0 m is considered for the simulation. An isotropic linear elastic constitutive model is employed, with parameters detailed in Table 1. The background mesh consists of cells sized 0.05 m × 0.05 m, with 4 material points in each mesh element, resulting in a total of 160 material points. Roller normal impermeable boundary is applied to the lateral surfaces, while the bottom is fully fixed and impermeable. The top surface of the column is permeable, allowing fluid to flow out through it. The initial conditions include an excess pore pressure $p_0 = 10$ kPa and zero effective stress. Not considering gravity, the consolidation process begins by applying a 10 kPa traction to the top material point layer and keeping it constant during the calculation. The time step is set to be $1.0 \times 10^{-5}$ s with the total simulation time of 2.0 s.

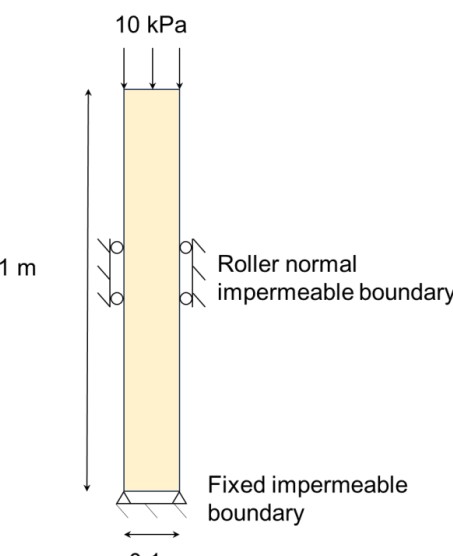

**Figure 4.** Schematic of the one-dimensional consolidation.



**Table 1** Material parameters for the one-dimensional consolidation

| Parameter | Value |
|---|---|
| Solid grain density $\rho_s$ (kg·m$^{-3}$) | 2650 |
| Young's modulus $E$ (MPa) | 10 |
| Poisson's ratio $\upsilon$ | 0.0 |
| Fluid density $\rho_w$ (kg·m$^{-3}$) | 1000 |
| Initial porosity $n$ | 0.3 |
| Bulk modulus of fluid $K_f$ (Gpa) | 2.2 |
| Hydraulic conductivity $K_s$ (m·s$^{-1}$) | 0.001 |


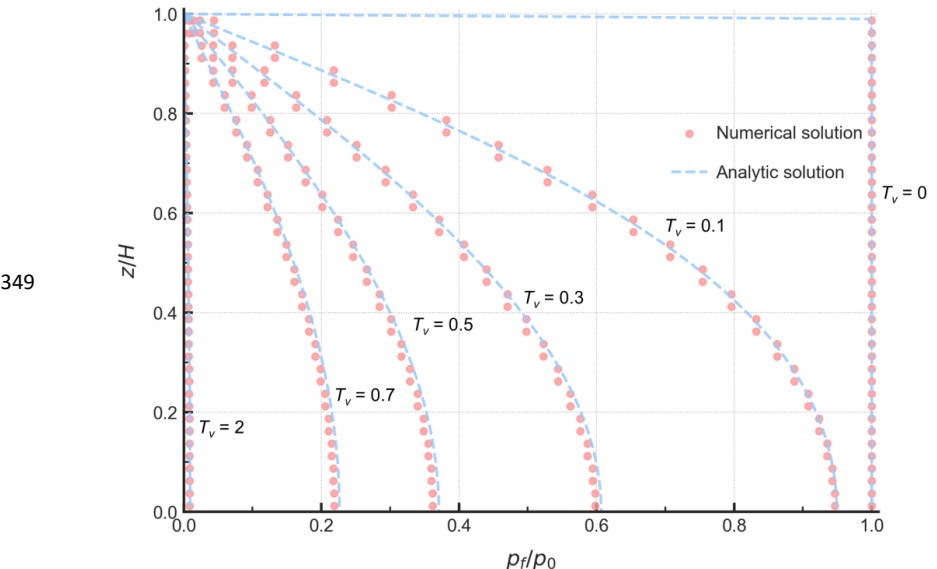

**Figure 5.** Comparison of pore pressure profiles from the proposed formulation with
351                              Terzaghi's solution.



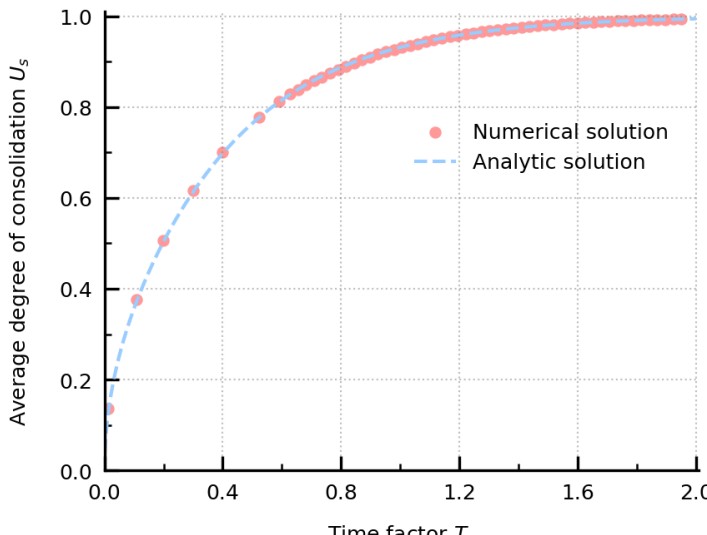

**Figure 6.** Comparison of the average degree of consolidation from the proposed
formulation with Terzaghi's solution.
Under such a constant loading, the deformation of the column is very small and
Terzaghi's one-dimensional consolidation theory is applicable. Fig. 5 presents a
comparison of the normalized pore pressure distribution at different time factors
between the numerical solution and the analytical solution (the time factor $T_v = C_v t /$
$H^2$, where $C_v$ is the coefficient of consolidation and $H$ is the drainage path length).
Initially, the pore pressure equals the external load, with the fluid phase undertaking
the external loading. Since the external loading is constant, the pore fluid is gradually
discharged from the top surface and the pore pressure begin to dissipate progressively
from the top. The numerical results show excellent agreement with the analytical
solutions, effectively capturing the dissipation process of the excess pore pressure
during consolidation. Additionally, the comparison of the average consolidation
degree (defined by strain) is presented in Fig. 6, indicating that the numerical results
accurately replicate the deformation process as the analytical solution shows.
**4.1.2 Large deformation**
For the large deformation condition, the same geometry and discretization as in the



small deformation case are used. However, a larger top traction (0.2 MPa) is applied
and a softer material ($E$ = 1MPa) is considered, and the hydraulic conductivity $K_s$ is
adjusted to be 0.0001 m·s⁻¹. Accordingly, the pore pressure is initialized at 0.2 MPa,
ensuring that the loading is initially fully carried by the fluid phase. Similar to the
small deformation case, after applying the constant loading, the pore pressure will
gradually dissipate, but now this process will generate considerable vertical
deformation. The decrease of the column-length is not negligible, therefore the small-
strain Terzaghi's theory is no longer applicable. Based on the large deformation
analytical solution (Xie and Leo, 2004), the evolution of pore pressure, top settlement
and the average degree of consolidation (defined by strain) can be expressed as,
$$p_f(z,t) = \frac{1}{m_{vl}} \ln[1 + (e^{m_{vl}p_a} - 1)\sum_{m=1}^{\infty}\frac{2}{M}\sin(\frac{Mz}{H})e^{-M^2T_v}] \tag{35}$$

$$S_t = H_0(1 - e^{-m_{vl}p_a})(1 - \sum_{m=1}^{\infty}\frac{2}{M^2}e^{-M^2T_v}) \tag{36}$$

$$U_s = 1 - \sum_{m=0}^{\infty}\frac{2}{M^2}e^{-M^2T_v} \tag{37}$$

where $m_{vl}$ = 1 / $E$ is the one-dimensional compressibility, $p_a$ is applied external load,
$H_0$ is the initial depth of the column, $z$ is the distance to the top surface. With the same
time step, the total simulation time is 300.0 s.

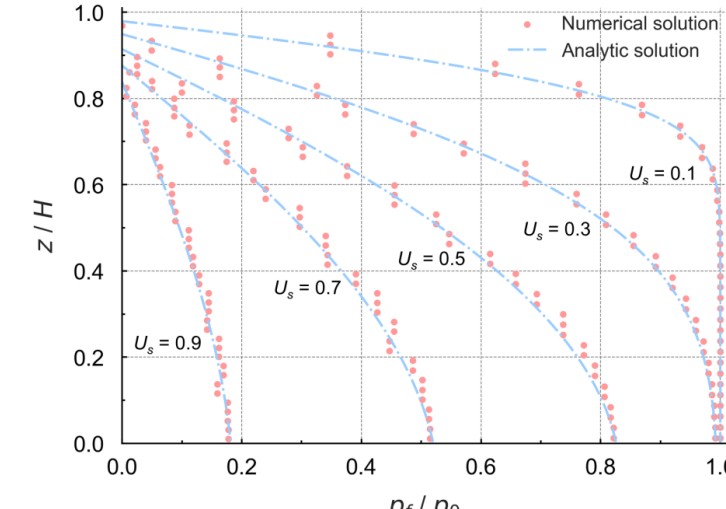


**Figure 7.** Comparison of pore pressure profiles from the proposed formulation with

388                          analytic solution.







**Figure 8.** Comparison of the top settlement from the proposed formulation with analytic solution.

Fig. 7 shows the numerical solution of pore pressure evolution along the column height against the results from the analytic solution at different average degrees of consolidation. In the small deformation case, the consolidation coefficient $C_v$ is equal to 1. While for the large deformation case, the consolidation coefficient $C_v$ is very small, so the consolidation is a long process. Hence, the pore pressure dissipation here is much slower than that in the small deformation case. The comparison shows that the numerical results are consistent with the analytic solutions and accurately depict this large deformation consolidation process. The cell average method used in the strain smoothing method will give the same volumetric strain rate for the particles in the same mesh cell, resulting in the same pore pressure distribution in each mesh cell, but the overall trend of this large consolidation process can still be captured. And Fig. 8 shows the evolution of the settlement at the top surface. The numerical result (final top settlement: 0.1815 m) is very close to the analytic result (final top settlement: 0.1802 m). The comparison demonstrates the validation and applicability of the proposed formulation in this two-phase large deformation process.



**4.2 Two-dimensional consolidation under local loading**
In this section, a two-dimensional elastic consolidation under a localized loading is
simulated, with the geometry and boundary conditions illustrated in Fig. 9. Due to the
symmetry of the problem, only half of the domain is modeled. The saturated material
domain possesses a dimension of 10.0 m × 10.0 m, while the background mesh
consists of cell elements sized 0.05 m × 0.05 m, with 4 material points in each cell
element, resulting in 1600 particles. Roller normal impermeable boundary is applied
to the lateral surfaces and the bottom, while the top surface is permeable and
unconstrained. Initially, a constant local loading of 20.0 kPa, spanning a width of 0.3
m, is applied on the left side of the top surface. Without considering gravity, the initial
stress and pore pressure are set to be zero. The isotropic linear elastic constitutive
model is used and the material parameters are provided in Table 2. The time step of
the simulation is $2.0 \times 10^{-4}$ s. The same simulation has been conducted in the previous
studies by semi-implicit MPM scheme (Yuan et al., 2023; Kularathna et al., 2021).
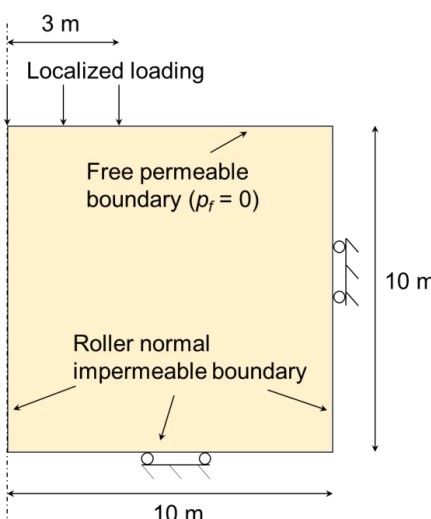
**Figure 9.** Model setup for the two-dimensional consolidation.






**Table 2** Material parameters for the two-dimensional consolidation

| Parameter | Value |
|---|---|
| Solid density $\rho_s$ (kg·m$^{-3}$) | 2700 |
| Young's modulus $E$ (MPa) | 10 |
| Poisson's ratio $v$ | 0.3 |
| Fluid density $\rho_w$ (kg·m$^{-3}$) | 1000 |
| Initial porosity $n$ | 0.3 |
| Bulk modulus of fluid $K_w$ (Gpa) | 2.2 |
| Hydraulic conductivity $K_s$ (m·s$^{-1}$) | 0.0001 |

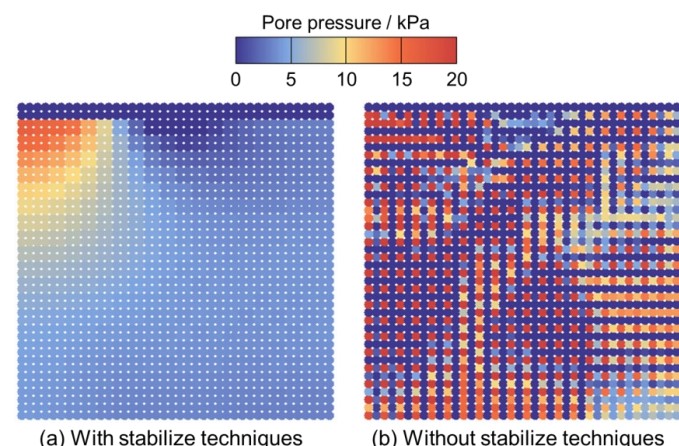

(a) With stabilize techniques        (b) Without stabilize techniques

**Figure 10.** Pore pressure distribution with stabilize techniques and without stabilize techniques at $t = 0.1$ s.

Fig. 10 illustrates the distribution of pore pressure at time $t = 0.1$ s, comparing results with and without stabilized techniques. In Fig. 10b, a spurious pore-pressure field with a checkerboard distribution is observed. In contrast, the result with stabilized techniques shows a smooth excess pore pressure field caused by the external loading (Fig. 10a). It demonstrates that the stabilized techniques can well mitigate pore pressure oscillation in the two phase MPM formulation, offering a stable pressure distribution. And the displacement distribution at $t = 0.1$ s is shown in Fig. 11. Consistent with the applied local loading, the displacement mainly occurs in the local loading region, indicating that the local loading is undertaken by the upper left corner area. The maximum displacement (6.737 mm) occurs at top left corner, which is





consistent with the result from the semi-implicit MPM formulation (Yuan et al., 2023).
Similar results are also obtained using the semi-implicit MPM with artificial
compressibility stabilization and fractional–step method (Yuan et al., 2023;
Kularathna et al., 2021). The stabilized techniques employed here can yield equivalent
results that are free of stress oscillations while accurately preserving the mechanical
behavior.

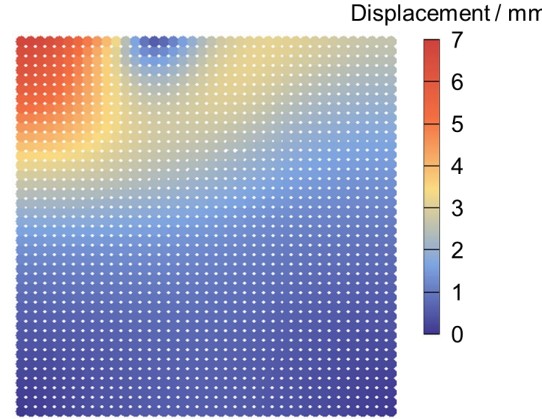

**Figure 11.** Distribution of displacement field at time $t = 0.1$ s.
**4.3 Cyclic loading test**
Inspirited by the lateral cycle loading test (Liang et al., 2023), we conduct a vertical
cyclic loading test of a saturated granular material. The model setup is shown in Fig.
12, where the saturated material is placed in a rigid box and subjected to a vertical
cyclic loading. The material domain measures 2 m in width and 1 m in height, and is
discretized by quadrilateral element with size of 0.05 m × 0.05 m. And there are 4
particles in each element, giving 3200 particles. Both the bottom and laterals are
normal impermeable and supported by rollers, and the top is unconstrained and
permeable. To apply a cycle loading, the top surface is prescribed by a sinusoidal
function periodic load of $40 sin 5\pi t$ kPa. Table 3 lists the material parameters used for
the isotropic linear elastic constitutive model. Before the cyclic stimulation, an
equilibrium condition is achieved by a linear gravity loading from 0 to 9.81 m/s$^2$
within $0 \leq t \leq 0.1$ s, and then the gravity remains constant. And to monitor the cycle



loading response, three monitoring points located at the bottle, middle and top of the
material domain (A, B, C) are selected (as shown in Fig. 12). The time step is set to be
$1.0 \times 10^{-5}$ s, and the simulation is terminated at 2.1 s.
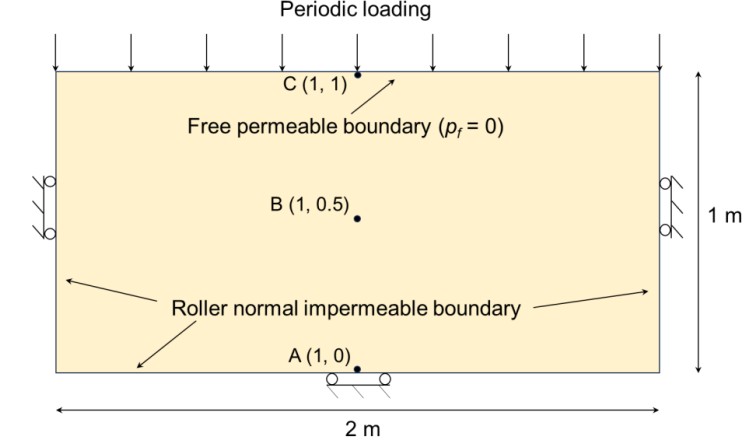
**Figure 12.** Schematic of cycle loading test.
**Table 3** Material parameters for the cycle loading test

| Parameter | Value |
|---|---|
| Solid density $\rho_s$ (kg·m$^{-3}$) | 2650 |
| Young's modulus $E$ (MPa) | 600 |
| Poisson's ratio $\upsilon$ | 0.3 |
| Fluid density $\rho_w$ (kg·m$^{-3}$) | 1000 |
| Initial porosity $n$ | 0.23 |
| Bulk modulus of fluid $K_w$ (Gpa) | 2.2 |
| Hydraulic conductivity $K_s$ (m·s$^{-1}$) | 0.001 |






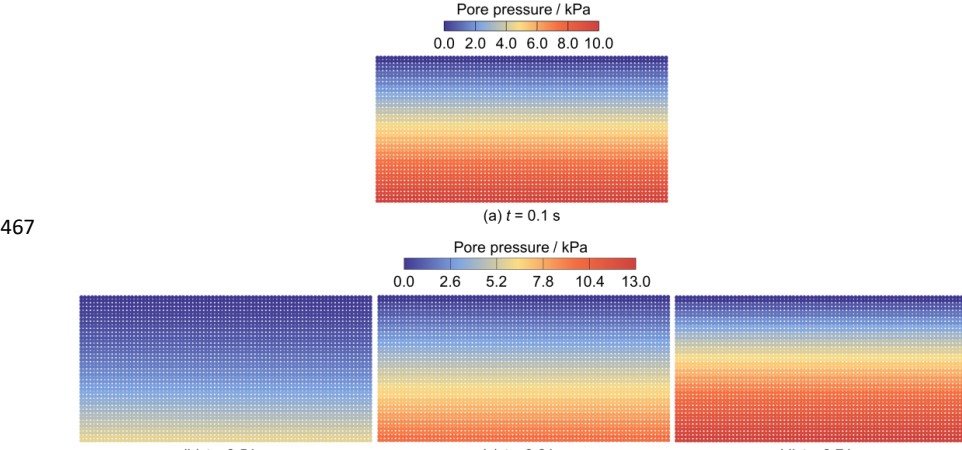

**Figure 13.** Distribution of pore pressure at $t$ = 0.1 s (hydrostatic Pressure), $t$ = 0.51 s, $t$
= 0.61 s, and $t$ = 0.71 s.
Fig. 13 shows the generated pore pressure at four different time instants. After the
application of linear gravity loading, an equilibrium condition is achieved and a
hydrostatic pressure field is generated (Fig. 13a). Subsequently, a vertical cyclic
loading is applied to the surface. When the material domain is subjected to
compressive loading, the pore pressure field increases, whereas under tensile loading,
the pore pressure field decreases correspondingly. This vertical cyclic shaking induces
an apparent periodic buildup and dissipation of excess pore pressure in the material
domain. In Fig. 13b, a clear pore pressure decrease due to tensile loading at $t$ = 0.51 s
can be seen. As the tensile loading gradually decreases and shifts into a compressive
loading, the pore pressure will gradually raise up. As a result, the pore pressure field
returns to the hydrostatic state at $t$ = 0.61 s (Fig. 13c). Subsequently, the compressive
loading leads to a further increase in pore pressure. As depicted in Fig. 13d, a
significant excess pore pressure field is regenerated. Therefore, the pore pressure in
the material domain exhibits periodic variations in response to the cyclic loading.
And to further present the cyclic dynamic response under the applied cyclic loading,
the evolution of pore pressure and displacement at the selected monitoring points is
presented in Fig. 14. The time history of pore pressure and displacement over time
demonstrates this cyclic loading response more quantitatively and vividly. The linear



gravity loading ends at $t = 0.1$ s, during which the displacement remains very small.
After that, the vertical loading will induce a relatively large displacement. Under the
sinusoidal periodic loading, the vertical displacement of point B and C exhibits a
sinusoidal variation, and the pore pressure at point A and B also changes accordingly.
It can be seen that these cyclic responses can be well captured by the proposed
stabilized MPM formulation.
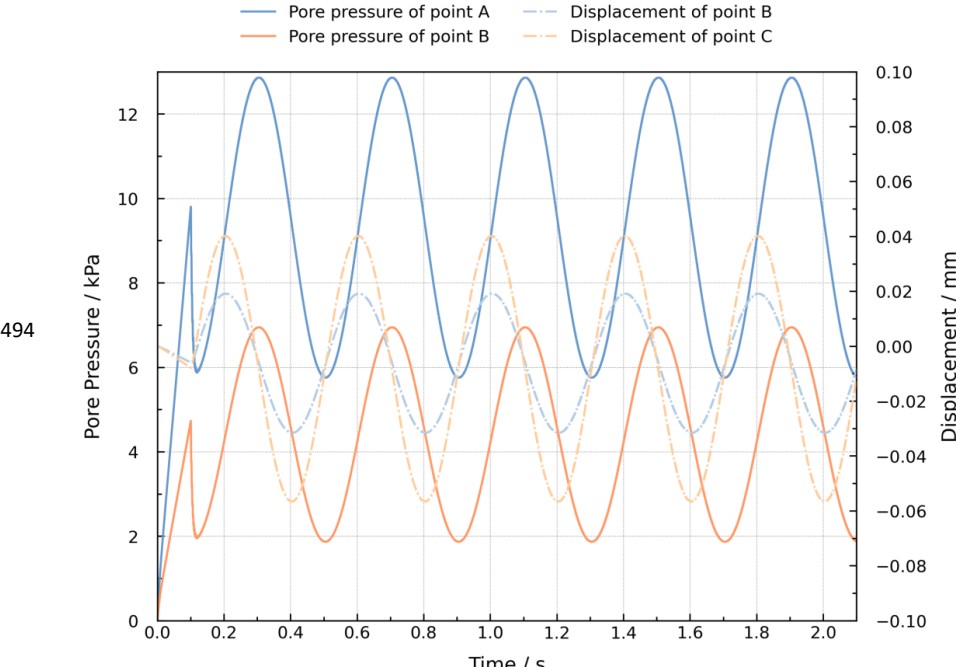
**Figure 14.** Evolution of pore pressure and displacement at selected points.
**4.4 Self-weight consolidation**
The large-deformation consolidation of an elastic slumping block under gravity
loading is presented in this section (Fig. 15), which is related to the settlement of a
very soft soil and has been simulated in previous studies (Zheng et al., 2021, 2022;
Sang et al., 2024; Wang et al., 2023a). The simulation focuses on the right half of a
symmetric domain with dimensions of 4 m width and 2 m height. The material
domain is discretized using quadrilateral element of size 0.125 m× 0.125 m, and 4
particles in each element, giving 1024 particles in total. No external load is applied,



making the consolidation process solely driven by the initial gravitational force at the
start of the simulation. The gravity linearly increases from 0 to 9.81 m/s$^2$ within $0 \le t$
$\le 0.1$ s and then remains constant. Both the top and right boundaries are unconstrained
and freely draining, while the left and bottom boundaries are normal impermeable and
supported by rollers. The gravity will give rise to pore pressure build-up, while the
deformation will lead to the dissipation of pore pressure over time. And two points ($P_1$,
$P_2$) at the bottle and middle are selected to evaluate the consolidation process (as
shown in Fig. 15). An isotropic linear elastic constitutive model is used and the
parameters are listed in Table 4. The total simulation time is 0.5, and the simulation is
performed with a time step equal to $1.0 \times 10^{-6}$ s.

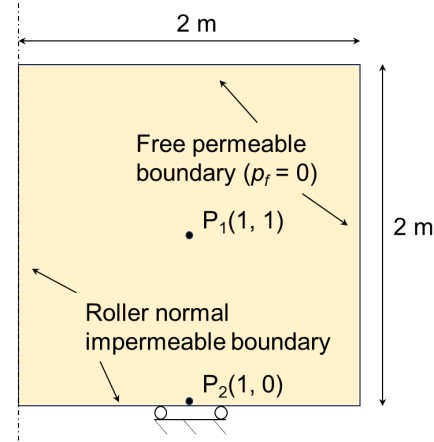

**Figure 15.** Schematic of the self-weight consolidation
**Table 4** Material parameters for the self-weight consolidation

| Parameter | Value |
|---|---|
| Solid density $\rho_s$ (kg·m$^{-3}$) | 2650 |
| Young's modulus $E$ (kPa) | 100 |
| Poisson's ratio $\upsilon$ | 0.3 |
| Fluid density $\rho_w$ (kg·m$^{-3}$) | 1000 |
| Initial porosity $n$ | 0.4 |
| Bulk modulus of fluid $K_w$ (Gpa) | 2.2 |
| Hydraulic conductivity $K_s$ (m·s$^{-1}$) | 0.0001 |



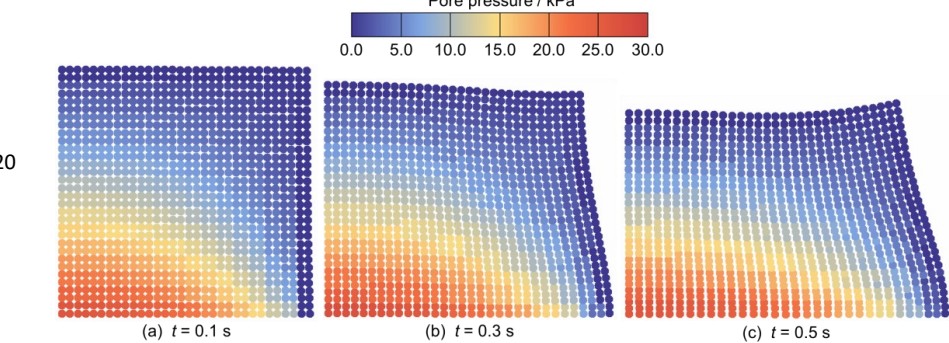

**Figure 16.** Pore pressures distribution at $t = 0.05$ s obtained with stabilize techniques

and without stabilize techniques.

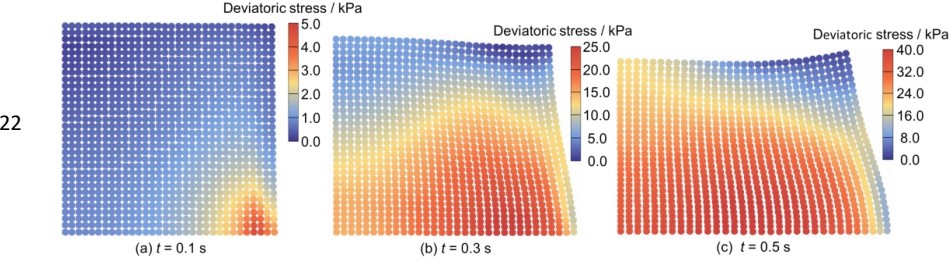

**Figure 17.** Pore pressures distribution at different times.

**Figure 18.** Deviatoric stress distribution at different times.

Initially, due to the relatively quick application of gravity loading, the pore fluid cannot be rapidly discharged and the loading process is carried out under approximately undrained condition. Therefore, the applied gravity loading will induce





excess pore pressure at the beginning. Fig. 16 shows pore pressure fields after gravity
loading ($t = 0.05$ s) with stabilized techniques and without stabilized techniques. One
can see that the result without stabilized techniques suffers from pore pressure
oscillations. The stabilized result, in contrast, eliminates spurious oscillations
effectively under the stringent undrained condition. Moreover, the distribution of pore
pressure and deviatoric stress at three different times (0.1 s, 0.3 s and 0.5 s) are
illustrated in Fig. 17 and 18, respectively. Upon the application of linear gravity
loading, a pore pressure field develops, gradually decreasing from the bottom left
corner upwards, as shown at $t = 0.1$ s (Fig. 17a). At this stage, the deformation is not
large, with a localized region of deviatoric stress distribution observed near the
bottom right corner (Fig. 18a). Subsequently, gravity continues to generate pore
pressure, and the deviatoric stress gradually increases as deformation progresses. As
the deformation develops under gravity, the pore pressure first reaches the maximum
value and then dissipates because of the deformation and drainage at the boundary.
This process can be observed in Fig. 17b, Fig. 18b and Fig. 17c, Fig. 18c. Both the
pore pressure and deviatoric stress filed change continuously along the large
deformation process. The absence of checkerboard oscillations shows the stability of
the proposed stabilized formulation in capturing the mechanical behavior of the
slumping block during the consolidation process.
To further verify the accuracy of the results, the time evolution of the pore pressure at
two points ($P_1$, $P_2$ in Fig. 15) is shown in Fig. 19, and the results are compared with
those of Zheng et al. (Zheng et al., 2022) using implicit stabilized MPM formulation
and Sang et al. (Sang et al., 2024) using implicit coupled MPM formulation. During
the linear gravity loading, pore pressure increases linearly, followed by non-
monotonic dissipation due to the Mandel-Cryer effect. The curves obtained using the
proposed stabilized formulation agree well with those of Zheng et al. (Zheng et al.,
2022) and Sang et al. (Sang et al., 2024). And the final displacement field (Fig. 20)
closely matches the results reported in previous studies (Wang et al., 2023a; Yuan et
al., 2023).



**Figure 19.** Pore pressures evolution at two selected points.

**Figure 20.** Displacement distribution at 0.5 s.

## 5 Discussion and conclusion

This study presents an explicit stabilized two-phase material point method for hydromechanical coupling problems in solid-fluid porous media. By incorporating the strain smoothing method and the multi-field variational principle in the single-point two phase MPM scheme, the proposed formulation effectively mitigates pore pressure oscillation and maintains numerical stability. The proposed two phase MPM formulation was initially validated through one-dimensional consolidation problem under both small and large deformation cases, with the numerical results showing



strong agreement with analytical solutions. Subsequently, the two-dimensional
consolidation under local loading and the cyclic loading test were performed,
demonstrating the formulation's robust capability to accurately capture dynamic
responses to external loading. Finally, the self-wight consolidation was analyzed to
showcase its efficacy in simulating both undrained and drained conditions, as well as
handling large deformation challenges. The proposed formulation produced results
that aligned closely with analytical solutions and outcomes from other approaches.
Particularly, the pore pressure instabilities were greatly mitigated by the stabilized
techniques, as clearly validated by the numerical results in terms of pore pressure.
With its effective and easy implemented stabilized techniques, the proposed
formulation is well-suited for analyzing a wide range of hydromechanical processes
under various undrained, drained, and loading conditions. It offers an effective and
reliable approach for simulating both static and dynamic processes in solid-fluid
porous media. This work is currently limited to linear elastic behavior of the solid
phase. Future efforts will focus on the practice and application involving more
complex large deformation problems and advanced constitutive models.
**Code and data availability**. The model developed in this study is based on the open
source MPM code, which is available on Github:
https://github.com/xzhang66/MPM3D-F90 (Zhang et al., 2016). The current version
of model is available from the project website: https://zenodo.org/records/14899281
under the Creative Commons Attribution licence. The exact version of the model used
to produce the results used in this paper is archived on Zenodo (Tang, 2025).
**Author Contributions.** XT developed the model and wrote the original draft of the
paper. WL, SH, LZ and MJ supervised the early stages of the study and provided
guidance. HZ, YS and ZH have actively contributed to the formal analysis, as well as
the writing and review of the paper. All authors were actively involved in the writing
process.
**Competing interest.** There are no known conflicts of interest associated with this
publication.



**Financial support.** This work was supported by the National Key Research and
Development program of China (Project No. 2022YFF0800604), the Major Program
of the National Natural Science Foundation of China (Grant No. 42090051), and the
China Scholarship Council (202304910567).

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
