# Peer review of "Stabilized two-phase material point method for hydromechanical coupling problems in solid-fluid porous media"

_EGUsphere, 2025_

## Author Comment (AC1)

Dear reviewer,

Thank you for your valuable suggestions for our manuscript, and we greatly appreciate the insightful comments. We believe these suggestions will greatly help us improve the quality of our manuscript. All changes in the revised manuscript are highlighted in red. Below, we address each comment in detail:

**(1) About the research background, the introduction could better emphasize the practical importance of simulating hydromechanical coupling in porous media to further highlight the motivation of the study.**

**Response:** Thank you for this suggestion. We have revised the first paragraph in the introduction to further highlight the motivation of the study. The revised version provides a more detailed explanation of our research motivation and importance.

**(2) In section 4.3, the total simulation time is not illustrated.**

**Response:** Sorry for this mistake, the total simulation time of example 2 is 0.1s, and we have added this content in the revised manuscript.

**(3) The discussion and conclusions could be strengthened by providing more detailed findings to highlight the work, including a brief discussion of how the proposed method compares to other stabilization techniques.**

**Response:** Thank you for the suggestion. We have revised the section of discussion and conclusion to provide more detailed findings that emphasize the significance of our work. A more detailed discussion is provided to show the strength of our formulation. We hope these revisions address your concern and enhance the clarity of our conclusions.

**(4) The figures and tables are generally clear but could be improved with clearer colors and better labeling and annotations.**

**Response:** Thanks for the suggestion. We have made some adjustments no the figures and tables in the manuscript, especially Fig. 13, 17 and 18. The modification makes the figures and tables clearer and more understandable.

**(5) A thorough review to ensure consistent and accurate terminology in the manuscript. Some terms, such as "pore pressure increasement," are non-standard and could be replaced with more conventional phrasing (e.g., "pore pressure increment").**

**Response:** Really thanks for this suggestion. About the language of the manuscript, we have conducted a comprehensive review to improve the quality of the manuscript and make sure that the language meets the requirement of scientific writing.

---

## Author Comment (AC2)

Dear reviewer,

Thank you for your valuable feedback on our manuscript. We truly appreciate your insightful comments, which we believe will significantly enhance the quality of our work. All changes in the revised manuscript are highlighted in red. And below are our replies to the comments point-by-point:

**(1) The language of the manuscript needs to be further improved for clarity, coherence and linguistic accuracy (like, line 107 "easy implemented", line 571 "the self-wight consolidation"), a thorough proofreading is recommended to eliminate any grammatical errors, spelling mistakes, or inconsistencies in terminology.**

**Response:** Thanks for your suggestion. We have conducted a comprehensive review to improve the language of the manuscript to eliminate grammatical error, spelling mistake and inconsistency in terminology. Hoping that the revised manuscript meets the standard of scientific writing.

**(2) In section 4.2, how long is the simulation time?**

**Response:** Sorry for the mistake, the simulation time of example 2 is 0.1s, and we have added it in the revised manuscript.

**(3) The introduction provides useful background information, but the specific research gap addressed by this study could be more clearly stated to show the research motivation.**

**Response:** Thank you for this suggestion. We have revised the first paragraph of the introduction to further emphasize the motivation behind this study. The revised version provides a clearer understanding of the rationale behind our work and how our study is motivated.

**(4) The discussion and conclusion section could be refined to strengthen the connection**

**between results and broader implications, clarify the limitations and highlight the significance of the findings.**

**Response:** Thank you for the suggestion. We have refined the section of discussion and conclusion to better establish the connection between our results and their broader implications, and provide more detailed findings that emphasize the significance of our work. Additionally, we have also clarified the limitations of our study. We believe these revisions effectively address your concerns and improve the clarity and depth of our conclusions.

Best,

Xiong (on behalf of all co-authors)